# Comparative Study of the Effects of GLP1 Analog and SGLT2 Inhibitor against Diabetic Cardiomyopathy in Type 2 Diabetic Rats: Possible Underlying Mechanisms

**DOI:** 10.3390/biomedicines8030043

**Published:** 2020-02-25

**Authors:** Abdelaziz M. Hussein, Elsayed A. Eid, Medhat Taha, Rami M. Elshazli, Raouf Fekry Bedir, Lashin Saad Lashin

**Affiliations:** 1Department of Medical Physiology, Mansoura Faculty of Medicine, Mansoura 35516, Egypt; lashinsaad@yahoo.com; 2Department of Internal Medicine and Endocrinology, Delta University for Science and Technology, Gamasa 35712, Egypt; sayedeid92@yahoo.com; 3Department of Anatomy, Mansoura Faculty of Medicine, Mansoura 35516, Egypt; medhattaha53@yahoo.com (M.T.); raouf_bedeer@yahoo.com (R.F.B.); 4Department of Biochemistry, Faculty of Physical Therapy, Horus University-Egypt, New Damietta 34518, Egypt; relshazly@horus.edu.eg; 5Department of Medical Physiology, Horus University, Damietta 34518, Egypt

**Keywords:** type 2 DM, cardiomyopathy, oxidative stress, caspase-3, TGF-β, TNF-α and tyrosine hydroxylase, SGLT2i, GLP1, norepinephrine

## Abstract

The present study investigated the possible cardioprotective effects of GLP1 and SGLT2i against diabetic cardiomyopathy (DCM) in type 2 diabetic rats and the possible underlying mechanisms. Methods: Thirty-two male Sprague Dawley rats were randomly subdivided into 4 equal groups: (a) control group, (b) DM group, type 2 diabetic rats with saline daily for 4 weeks, (c) DM + GLP1, as DM group with GLP1 analogue (liraglutide) at a dose of 75 µg/kg for 4 weeks, and (d) DM + SGLT2i as DM group with SGLT2 inhibitor (dapagliflozin) at a dose of 1 mg/kg for 4 weeks. By the end of treatment (4 weeks), serum blood glucose, homeostasis model assessment insulin resistance (HOMA-IR), insulin, and cardiac enzymes (LDH, CK-MB) were measured. Also, the cardiac histopathology, myocardial oxidative stress markers (malondialdehyde (MDA), glutathione (GSH) and CAT) and norepinephrine (NE), myocardial fibrosis, the expression of caspase-3, TGF-β, TNF-α, and tyrosine hydroxylase (TH) in myocardial tissues were measured. Results: T2DM caused significant increase in serum glucose, HOMA-IR, serum CK-MB, and LDH (*p* < 0.05). Also, DM caused significant myocardial damage and fibrosis; elevation of myocardial MDA; NE with upregulation of myocardial caspase-3, TNF-α, TGF-β, and TH; and significant decrease in serum insulin and myocardial GSH and CAT (*p* < 0.05). Administration of either GLP1 analog or SGLT2i caused a significant improvement in all studied parameters (*p* < 0.05). Conclusion: We concluded that both GLP1 and SGLT2i exhibited cardioprotective effects against DCM in T2DM, with the upper hand for SGLT2i. This might be due to attenuation of fibrosis, oxidative stress, apoptosis (caspase-3), sympathetic nerve activity, and inflammatory cytokines (TNF-α and TGF-β).

## 1. Introduction

Diabetes mellitus (DM) is a systemic metabolic disease, which is characterized by chronic hyperglycemia due to lack of insulin secretion, insulin action, or both [1,2]. In 2015, an epidemiological study demonstrated that 415 million adults are living with DM, and this number is projected to reach 642 million by 2040 [3]. Type 2 DM (T2DM) is commonly associated with several cardiovascular complications such as coronary artery disease (CAD), stroke, and diabetic cardiomyopathy (DCM), which account for up to 65% of all deaths in diabetic patients [4]. DCM is defined as the presence of heart failure in absence of coronary artery disease [5]. DCM is associated with systolic and diastolic functions in which the prognosis is poor with an annual mortality of 15%–20%, reviewed in [6]. The pathophysiological mechanisms underlying DCM include excessive reactive oxygen species (ROS) production, apoptosis, overactivity of myocardial renin angiotensin system (RAS) and insulin-like growth factor-1 (IGF-1) as well as transforming growth factor beta 1 (TGF-β1), collagen deposition and fibrosis [7,8]. Therefore, strategies aiming to modulate fibrosis, inflammation, apoptosis and oxidative stress could be effective in management of DCM. Also, a recent study reported a significant increase in sympathetic nerve endings density (by measuring tyrosine hydroxylase activity) in the hearts of type 2 diabetic rats [9].

A novel class of medication called sodium/glucose cotransporter 2 inhibitors (SGLT2i) are used nowadays in the treatment of T2DM. Several experimental studies demonstrated the beneficial cardiovascular effects of SGLT2i independent of their anti-hyperglycemic actions [10,11,12,13,14]. Also, glucagon-like peptide 1 (GLP-1) is a gut-derived incretin hormone that stimulates insulin and suppresses glucagon secretion, inhibits gastric emptying, and reduces appetite and food intake. Therapeutic approaches for enhancing incretin action include degradation-resistant GLP-1 receptor agonists (incretin mimetics) and inhibitors of dipeptidyl peptidase-4 (DPP-4) activity (incretin enhancers). Clinical trials with the incretin mimetic show reductions in fasting, postprandial glucose concentrations and hemoglobin A_1c_ (HbA_1c_) [15]. In the literature, no previous studies, to the best of our knowledge, investigated the effects of these new lines of antidiabetic agents against the diabetic cardiomyopathy and cardiac autonomic neuropathy (CAN) in type 2 DM. So, the present study was designed to investigate the effects of the GLP1 analogue and SGLT2 inhibitor on DCM and diabetic CAN in T2DM and to clarify their underlying mechanisms.

## 2. Materials and Methods

### 2.1. Experimental Animals

Thirty-two male Sprague Dawley rats, aged 10–12 weeks, weighing 190 ± 10 g, were housed in the medical experimental research center (MERC) at Mansoura Faculty of Medicine in separate cages under controlled environmental conditions (12 h light/dark cycle and temperature 24 °C). Rats had a free access to water. All protocols of the present study were approved by the institutional review board (IRB) of Mansoura Faculty of Medicine (approval code # = R/19.02.421, 30 March 2019).

### 2.2. Study Design

Rats were randomly subdivided into 4 equal groups (8 rats each) as follows:Normal control group—including normal nondiabetic rats that received saline 0.5 mL via gastric gavage and 0.5 mL saline via subcutaneous (S.C.) injection;Diabetes Mellitus (DM) group—including T2DM rats that received saline 0.5 mL via gastric gavage and 0.5 mL via S.C. saline injection;DM + GLP1 group—including T2DM rats that received GLP1 analogue (liraglutide) at a dose of 75 µg/kg dissolved in 0.5 mL saline via S.C. injection and 0.5 mL saline via gastric gavage once daily for 4 weeks [16];DM + SGLT2i group—including T2DM rats that received SGLT2 inhibitor (dapagliflozin) at a dose of 1 mg/kg dissolved in 0.5 mL saline via oral gastric gavage and 0.5 mL via S.C. saline injection daily for 4 weeks [17].

### 2.3. Type 2 DM Rat Model

The details of induction of Type 2 DM in rats was mentioned in our previous study [18]. Briefly, rats were given a high-fat diet for 4 weeks. Then, a single dose of streptozotocin (STZ) (35 mg/Kg) was injected intraperitoneally (i.p.), and the concentration of blood glucose was measured within 48 h. Rats with blood glucose > 200 mg/dL were included in the present study.

### 2.4. Measurement of Serum Glucose, Insulin and Cardiac Enzymes and Calculation of HOMA Index

Fasting blood sugar and insulin were measured using commercially available kits (SPIN REACT, Spain for blood sugar and ELISA insulin kits, Sun-Red biology and technology, Shanghai #cat no 201-11-0708 for insulin) according to the manufacturer’s instructions. The homeostasis model assessment (HOMA) insulin resistance (IR) index was calculated from fasting insulin and fasting blood glucose using Mathew’s formula [19].

The serum levels of creatine kinase (CK)-MB and lactate dehydrogenase (LDH) were measured using commercially available kits (bioMérieux Diagnostics, Milan, Italy for CK-MB and Bayer Diagnostics Ltd., Baroda, India for LDH) according to the manufacturer’s instructions.

### 2.5. Measurement of Myocardial Oxidative Stress Markers (MDA, GSH and Catalase Activity)

A piece of left ventricle (about 50–100 mg) was dissected and homogenized in cold buffer (50 mM potassium phosphate, pH 7.5, 1 mM EDTA) using a mortar and pestle. The tissue homogenate was centrifuged at 4000 rpm for 15 min at 4 °C. Then, the supernatant was collected and stored at −20 °C until time of biochemical assay. The concentrations of malondialdehyde (MDA) and reduced glutathione (GSH) and the catalase enzyme activity were measured in the supernatant using a colorimetric method according to the manufacturer’s instructions (Bio-Diagnostics, Dokki, Giza, Egypt).

### 2.6. Real-Time PCR for the mRNA of TNF-α Gene Expression in Heart Tissues

A piece of left ventricle (50–100 mg) was homogenized in 1 mL of Trizol to obtain total RNA according to the manufacturer’s instructions (Invitrogen Corporation, Grand Island, New York, USA). Reverse transcription was done using 1 µg total RNA and a cDNA kit (high-capacity cDNA archive kit). The sequence of the primers of the tested genes were TNF-α: TNFα (295 bp) forward 5′-TACTGAACTTCGGGGTGATTGGTCC-3′, reverse 5′-CAGCCTTGTCCCTTGAAGAGAACC-3′; and GAPDH forward: 5′-TATCGGACGCCTGGTTAC-3′, reverse: 5′-CTGTGCCGTTGAACTTGC-3′. The details of PCR reaction and data analysis for the expression of nrf2 were mentioned in our previous work [20].

### 2.7. Histopathological Examination of the Heart Tissues by H&E and Masson Trichrome

The left ventricle of the heart was dissected and fixed in 10% neutral buffered formalin, then heart tissues were embedded in paraffin and sectioned at 3 μm thickness and stained using hematoxylin and eosin (H&E). Slides were examined for signs of cardiomyopathy, which included deranged myocardial cells, uneven cytoplasm distribution, edema, inflammatory cells infiltrate, hypertrophy of muscle fibers, rupture of myocardial fibers and irregular nuclei [21], using light microscope (Leica_DM500 with camera Leica_ICC50HD) with Camera software LEICA Application Suite (LAS) EZ, Version 3.1.1 (Physiology and Biotechnology Laboratory, Department of Animal Production, Faculty of Agriculture, Mansoura University). Also, slides stained with Masson trichrome were examined for interstitial fibrosis and collagen deposition, which appeared blue in color.

### 2.8. Immunohistochemical Examination for TGF-beta, caspase-3 and Tyrosine Hydroxylase

The tissue section was deparaffinized, rehydrated, washed, immersed in 3% hydrogen peroxide and then digested with pepsin for antigen retrieval. After the blocking of unspecific binding by serum, the section was incubated with primary antibodies of anti-caspase-3 (Abcam, cat#: ab79123), dilution 1:1000; mouse monoclonal antibody to TGF-β, dilution 3:1000 (DAKO); and TH (rabbit polyclonal anti-rat TH antibodies, 1:100, ab11370, Abcam, Boston, Massachusetts, USA) at 4 °C overnight. Diaminobenzidine/peroxidase substrate was used to produce a brown-colored signal. The section was counterstained, dehydrated, cleared and coverslipped. Phosphate buffered saline (PBS) was used to replace primary antibody, and adjacent sections were used as negative control. TH was quantified as the percent of myocardial area (region of interest, ROI) occupied by positive staining (calculated by averaging the values from ten fields at 10× magnification) for each left ventricular area by using image J software).

### 2.9. Statistical Analysis

Data processing and analysis was done by SPSS (statistical package of social science) version 17 (IBM compatible PC, Microsoft and SPSS/PC package version, USA). One-way ANOVA with Tukey’s post hoc test was used. *p* < 0.05 was considered significant.

## 3. Results

### 3.1. Effects of SGLT2i and GLP1 on Blood Glucose, Insulin, HOMA-IR, LDH and CK-MB in T2DM

Compared to the normal control group, the levels of blood glucose, HOMA, LDH and CK-MB were significantly higher in the DM group, while the level of insulin was significantly lower in the DM group (*p* < 0.05). On the other hand, the levels of blood glucose, HOMA, LDH and CK-MB were significantly attenuated in DM + SGLT2i and DM + GLP1 groups compared to the DM group, while insulin was significantly increased in treated groups compared to the DM group (*p* ˂ 0.05). Moreover, there were no statistically significant differences between DM + SGLT2i and GLP1 groups in these parameters, except blood glucose, which was significantly lower in the DM + SGLT2i group than the DM + GLPT1 group (Table 1).

### 3.2. Effects of SGLT2i and GLP1 on Markers of Oxidative Stress (MDA, GSH and CAT) In Heart Tissues

The level of MDA in myocardial tissues showed a significant increase in the DM group compared to that of the normal control group (*p* ˂ 0.001) and a significant decrease in treated groups (DM + SGLT2i and DM + GLP1 groups) compared to that of the DM group (*p* ˂ 0.01). Also, MDA levels were significantly lower in the DM + SGLT2i group than DM + GLP1 group (Figure 1A). On the other hand, the levels of the antioxidants (GSH and CAT) were significantly lower in the DM group compared to that of the normal control group (*p* ˂ 0.001) with significant increase in their levels in treated groups (DM + SGLT2i and DM + GLP1 groups) compared to the DM group (*p* ˂ 0.01). There was no statistically significant difference in the CAT activity between DM + SGLT2i and DM + GLP1 groups, while the concentration of GSH was significantly higher in DM + SGLT2i than that in the DM + GLP1 group (Figure 1B,C).

### 3.3. Effects of SGLT2i and GLP1 on the Proinflammatory Cytokine (TNF-α) mRNA in Heart Tissues

The expression of TNF-α mRNA in heart tissues was significantly higher in DM group than normal control group (*p* < 0.001). This elevation in TNF-α expression was significantly attenuated in treated groups (DM + SGLT2i and DM + GLP1 groups) compared to the DM group (*p* < 0.01). Moreover, the DM + SGLT2i group showed significant reduction in TNF-α expression compared to that of the DM + GLP1 group (*p* < 0.01) (Figure 2).

### 3.4. Effects of SGLT2i and GLP1 on Myocardial Morphology and Fibrosis

H&E histopathological examination revealed normal myocardial architecture with regular arrangement of myocardial fibers and nuclei in the normal control group (Figure 3A), while hearts obtained from the DM group showed irregular arrangement of fibers and nuclei with wide spacing of the cardiomyocytes, interstitial edema and hemorrhage and myocardial necrosis (Figure 3B). On the other hand, heart specimens from DM + SGLT2i and DM + GLP1 groups showed regular arrangement of cardiomyocytes with minimal deformed nuclei (Figure 3C,D).

Masson trichrome examination of the heart tissues for fibrosis revealed that no collagen fibers and fibrosis in heart specimens were obtained from the normal control group (Figure 4A), while hearts of the DM group showed excess deposition of collagen fibers in heart tissues (Figure 4B). Also, heart tissues obtained from DM + SGLT2i and DM + GLP1 groups revealed moderate interstitial fibrosis (Figure 4C,D).

### 3.5. Effects of SGLT2i and GLP1 on Myocardial Norepinephrine and Tyrosine Hydroxylase

The concentration of myocardial norepinephrine and TH density were significantly higher in the DM group than that of the normal control group (*p* < 0.001). This elevation was significantly attenuated in DM + SGLT2i and DM + GLP1 groups compared to the DM group (*p* < 0.01). Also, the level of NE and the density of TH were significantly lower in the DM + SGLT2i group than that in the DM + GLP1 group (*p* < 0.01) (Figure 5A,B). The specimens of the heart tissues from the normal group showed evenly distributed TH-positive nerve fibers (Figure 5C). Hearts of the DM group showed a large number of TH-positive small nerve branches, which had a “snow-like” distribution (Figure 5D). Also, heart specimens of rats in DM + SGLT2i and DM + GLP1 groups showed TH-positive sympathetic fibers (Figure 5E,F).

### 3.6. Effects of SGLT2i and GLP1 on Apoptotic Marker (caspase-3) in Myocardial Tissues

Examination of the expression of caspase-3 by immunostaining revealed significant increase in its expression in DM groups compared to that of the normal control group (*p* ≤ 0.001). In addition, its expression was significantly attenuated in treated groups (DM + SGLT2i and DM + GLP1) compared to that of the DM group (*p* ≤ 0.01). Moreover, its expression was significantly lower in the DM + SGLT2i group compared to that of the DM + GLP1 group (*p* ≤ 0.01) (Figure 6A). Heart specimens obtained from the normal control group showed negative cytoplasmic expression for caspase-3 (Figure 6B). Hearts obtained from the DM group showed marked cytoplasmic expression of caspase-3 in cardiomyocytes (Figure 6C). On the other hand, heart specimens obtained from the DM + SGLT2i showed minimal expression of caspase-3 in myocardial tissues (Figure 6D), while hearts obtained from the DM + GLP1 group showed moderate expression of caspase-3 in cardiomyocytes (Figure 6E).

### 3.7. Effects of SGLT2i and GLP1 on Inflammatory Cytokine (TGF-β) in Myocardial Tissues

Examination of the expression of TGF-β by immunostaining revealed significant increase in its expression in DM groups compared to that of the normal control group (*p* ≤ 0.001). In addition, its expression was significantly attenuated in treated groups (DM + SGLT2i and DM + GLP1) compared to that of the DM group (*p* ≤ 0.01). Moreover, its expression was significantly lower in the DM + SGLT2i group compared to that of the DM + GLP1 group (*p* ≤ 0.01) (Figure 7A). Heart specimens obtained from the normal control group showed negative expression for TGF-β (Figure 7B), while hearts obtained from the DM group showed marked interstitial expression of TGF-β in cardiomyocytes (Figure 7C). On the other hand, heart specimens obtained from the DM + SGLT2i group showed minimal expression of TGF-β in myocardial tissues (Figure 7D), while hearts obtained from the DM + GLP1 group showed moderate expression of TGF-β in the myocardium interstitium (Figure 7E).

## 4. Discussion

The main findings of the present study include the following: a) T2DM was characterized by significant elevation of blood glucose and cardiac enzymes, insulin resistance, disturbed myocardial morphology with myocardial fibrosis and oxidative stress. Also, b) T2DM was associated with up-regulation of apoptotic markers (caspase-3) and inflammatory cytokines (TGF-β) and increased sympathetic nerve density with higher myocardial NE, and c) pretreatment with either SGLT2i or GLP1 significantly improved the studied parameters, and SGLT2i offered a more protective effect than GLP1.

In the present study, a well-known rat model of T2DM (high-fat diet for one month followed by a single low dose of STZ) was adopted. The present study found high blood glucose with relative insulin deficiency and high HOMA-IR in the DM group, indicating T2DM development. Previous studies reported similar findings [18,22,23]. Lack of insulin and hyperglycemia developed in this animal model could be due to partial damage of pancreatic β-cells via the cytotoxic action of STZ and block of insulin receptors by a high-fat diet [22,23]. In the present study, we found that treatment with either SGLTi or GLP1 significantly improved the impaired glucose homeostasis and insulin resistance; however, the effect of SGLT2i was more powerful than that of the GLP1 analog on control of blood glucose and improvement of insulin resistance. These findings suggest these agents might improve blood glucose by increasing insulin secretion and glucose uptake by cells. A recent study by Li et al. [24] concluded that empagliflozin (one of the SGLT2 inhibitors) provided better glycemic control in T2DM.

The present study demonstrated significant cardiac damage in T2DM, as evidenced by the rise in the levels of cardiac enzymes (CK-MB and LDH) and by the morphological changes such as myocardial atrophy, degeneration, necrosis and inflammatory cell infiltrates observed in hearts of the DM group. The rise in cardiac enzymes in serum might be due to loss of the integrity of cardiomyocytes and release of cytosolic cardiac enzymes. Similar findings were demonstrated by other researchers [25,26,27]. Moreover, we found significant improvement in cardiac enzymes and morphology by both SGLTi and GLP1 treatments, and SGLT2i offered a more powerful cardioprotective action than GLP1. Consistent with these findings, Andreadou et al. [10] demonstrated a beneficial effect for empagliflozin (SGLT2i) against the myocardial ischemic injury at both vitro and in vivo levels. Also, Byrine et al. [12] demonstrated that empagliflozin improved the cardiac functions in an experimental model of pressure overload-induced heart failure. Moreover, Ye et al. [13] found that dapagliflozin attenuated cardiomyopathy in type 2 diabetic mice. Also, the GLP1 analogue demonstrated cardioprotective effects against myocardial infarction in mice [28] and cardiac remodeling in type 2 diabetic rats [29].

Pathogenesis of DCM is a complex and multifactorial process in which persistent hyperglycemia and insulin resistance lead to myocardial oxidative stress, inflammation, activation of the renin angiotensin aldosterone system (RAAS), autonomic neuropathy and myocardial apoptotic cell deaths. These molecular changes promote development of cardiac hypertrophy and fibrosis [30]. Activation of RAAS causes an increase in the activity of NADPH oxidase, which may directly promote cardiac fibrosis by activation of a pro-fibrotic TGF-β1/Smad 2/3 signaling pathway [31]. So, in the current study we investigated the roles of oxidative stress, inflammatory cytokines and apoptosis in the development of myocardial fibrosis and the effects of SGLT2i and GLP1 on these molecular mechanisms.

Oxidative stress plays an important role in the pathophysiology of diabetic cardiomyopathy (DCM). Hyperglycemia leads to excessive generation of free oxygen radicals, which promote the development of myocardial apoptosis and fibrosis. We found, in the current work, enhanced myocardial oxidative stress in diabetic rats as evidenced by a significant rise in MDA (marker of lipid peroxidation) and with significant decreases in CAT activity and the concentration of GSH in the hearts of the DM group. Yang et al. [26] and Wilson et al. [32] demonstrated similar findings, suggesting the important role of oxidative stress to DCM in T2DM. Moreover, the present study demonstrated significant improvement in myocardial oxidative stress by treatment with either SGLTi or GLP1, suggesting antioxidant actions for both agents. We suggest that the antioxidant actions of SGLTi and GLP1 might be a potential mechanism for their cardioprotective effects of type 2 diabetic cardiomyopathy in rats. The antioxidant actions of SGLT2i and GLP1 were reported in previous studies [33,34]. Li et al. [33] demonstrated that empagliflozin (SGLT2i) attenuated DCM in type 2 diabetic cardiomyopathy in mice via inhibition of oxidative stress generated by NADPH oxidases. Olgar and Turan [34] found that dapagliflozin had an antioxidant cardioprotective action via Zn^2+^ transporters, matrix metalloproteinases and oxidative stress. Also, they reported that liraglutide (GLP1) attenuated oxidative stress and fibrosis in cardiac tissues in mice treated with a high-fat diet.

Apoptotic cell damage is another mechanism for DCM, in which elevated blood sugar in DM promotes the release of mitochondrial cytochrome C to the cytosol, resulting in activation of caspase-3 and apoptotic cell death of cardiomyocytes [35]. Also, previous studies demonstrated the role of anti-apoptotic p53 protein [36] and proapoptotic Bcl-2 proteins [37] in DCM in diabetic hearts. In the current study, we found significant increase in caspase-3 in myocardial tissues of diabetic rats, suggesting involvement of apoptosis in the development of DCM. Also, we demonstrated significant attenuation in caspase-3 expression with GLP1 and SGLT2i treatment, suggesting that inhibition of apoptosis might be a mechanism for the cardioprotective effects of these agents against DCM. Moreover, SGLT2i exhibited more anti-apoptotic effects than GLP1 in DCM, as shown in our results. In agreement with our findings, an invitro study by Buteau et al. [38] demonstrated that treatment with GLP-1 activated the antiapoptotic proteins and attenuated the apoptotic process independent of oxidative stress.

Myocardial fibrosis and inflammation are key features in DCM [39,40]. The mechanisms of the inflammatory process in early DCM involves activation of inflammatory cytokines such as tumor necrosis factor-𝛼 (TNF𝛼), interleukins (IL-1𝛽, IL-6) and cellular adhesion molecules such as ICAM-1 and VCAM-1, as reviewed in [41]. In the current work we investigated the expression of TNF-α and found that its expression at the level of mRNA was increased in the heart tissues of type 2 diabetic rats, confirming its activation and role in DCM. Also, transforming growth factor (TGF)-β is involved in the differentiation of fibroblasts to myofibroblasts, and this causes excessive production of collagen and fibrosis [42]. We found a significant increase in the collagen fiber content by Masson trichrome and in the expression of TGF-beta in the heart tissues of the diabetic group, suggesting the possible role of TGF-beta in cardiac fibrosis. Consistent with this hypothesis, Martin et al. [43] reported that tranilast (inhibitor of TGF-beta) attenuated the deposition of cardiac matrix in diabetic hearts. On the other hand, [33] reported a significant increase in oxidative stress and collagen deposition in diabetic rat hearts without an increase in TGF-β expression. Moreover, the current work found that treatment with either SGLT2i or GLP1 caused a significant reduction in myocardial fibrosis and myocardial expression of TGF-β, suggesting their antifibrotic activity against DCM via suppression of TGF-β.

An in vitro study by Dorecka and colleagues [44] demonstrated that treatment with GLP-1 (100 nM) inhibited TNF-α-induced expression of receptors for advanced glycation end products (RAGE), ICAM-1 and VCAM-1 in cultured human retinal pigment epithelial cells, suggesting that attenuation of TNF-α might be a mechanism for the action of GLP1. In line with this conclusion, the present study found that TNF-α expression was attenuated at the level of mRNA in cardiac tissues by GLP1 treatment.

Also, in the present study we investigated the effects of GLP1 and SGLT2i on the sympathetic nerve fiber density and myocardial NE content. The current showed significant increase in the density and number of sympathetic fibers in diabetic hearts, as evidenced by the significant increase in TH density in the hearts of type 2 diabetic rats with higher levels of NE in the myocardium, suggesting sympathetic overactivity in heart tissues in early DM. In line with these findings, Otake et al. [45] demonstrated a higher density of sympathetic nerves in diabetic rats. Also, Bakovic et al. [46] demonstrated that the increased sympathetic nerve fiber density in diabetic hearts started 2 weeks after induction of DM and reaching its peak after 2 months, then declined 6 to 12 months after DM induction. Also, a recent study by Hussein and colleagues demonstrated similar findings [9]. Moreover, we found in the current work a significant reduction in TH density and myocardial NE contents by either SGLT2i or GLP1, suggesting attenuation of the sympathetic nerve fiber activity is another possible mechanism for the cardioprotective effects of these new antidiabetic agents in DCM. Also, the SGLT2i exhibited more ameliorating effects than GLP1 on the density of sympathetic nerve endings and myocardial NE contents.

## 5. Conclusions

T2DM might be associated with DCM, which is characterized by myocardial fibrosis and which might be due to enhanced oxidative stress, upregulation of inflammatory cytokines (TGF-β, TNF-α) and apoptotic proteins (caspase-3) and increased activity of myocardial sympathetic nerve fibers. Treatment of diabetic rats with either SGLT2i (dapagliflozin) or GLP1 (liraglutide) protected the hearts from DCM, and SGLT2i offered more cardioprotective effects than GLP1. The cardioprotective effects of both agents might be due to inhibition of oxidative stress, the apoptotic process, the inflammatory process, fibrosis, the sympathetic nerve fiber activity and myocardial NE.

## Figures and Tables

**Figure 1 biomedicines-08-00043-f001:**
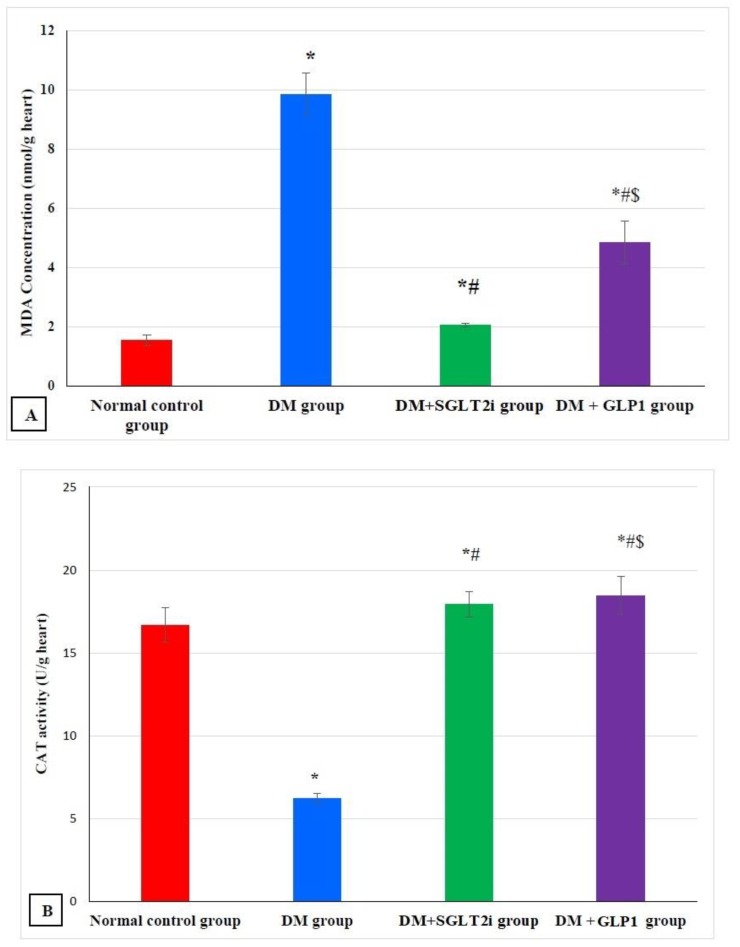
Myocardial oxidative stress markers in different groups. (**A**) Malondialdehyde (MDA) concentration (nmol/g heart tissues), (**B**) catalase enzyme activity (U/g heart tissues) and (**C**) reduced glutathione (GSH) concentration (mmol/g heart tissues). * Significant vs. control group, ^#^ significant vs. DM group, and ^$^ significant vs. DM + SGLT2i group.

**Figure 2 biomedicines-08-00043-f002:**
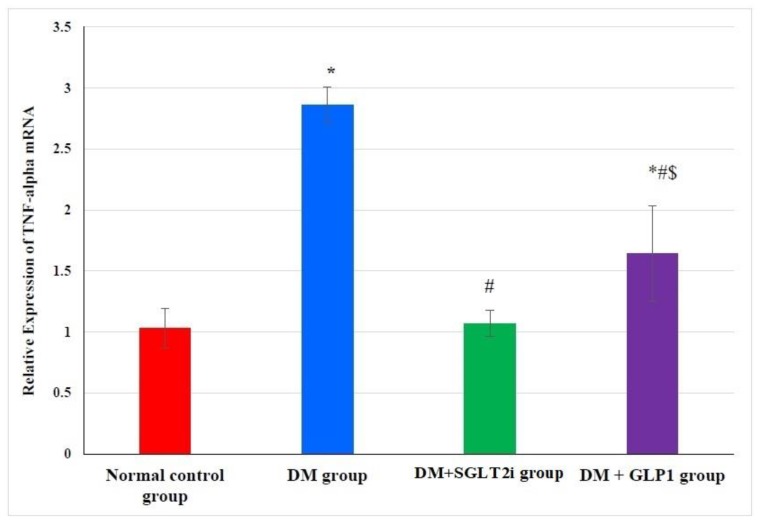
Relative expression of TNF-α in myocardial tissues at the level of mRNA in different groups by real-time PCR. * Significant vs. control group, ^#^ significant vs. DM group, and ^$^ significant vs. DM + SGLT2i group.

**Figure 3 biomedicines-08-00043-f003:**
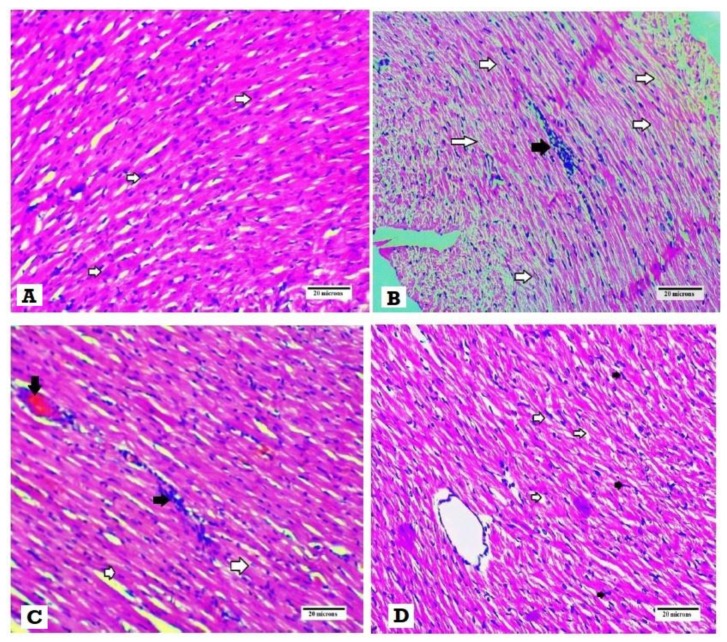
Histopathological examination of the heart tissues stained with H&E. (**A**) Heart specimen from control group showing regularly arranged cardiomyocytes and their nuclei (white arrows) (400×), (**B**) heart specimen from the DM group showing disarranged swollen cardiomyocytes with deformed nuclei of cardiomyocytes and wide myocardial gap (white arrows) and inflammatory cells infiltrate (black arrows) (400×), (**C**) heart specimen from the DM + GLP1 group showing minimal interstitial edema and deformed nuclei of cardiomyocytes )white arrows), inflammatory infiltrate (black arrows) and interstitial hemorrhage (black arrow) (400×), and (**D**) heart specimen from the DM + SGLT2i group showing regular arrangement of cardiomyocytes with regular arranged nuclei (black arrow) with minimal widening of myocardial gap (400×).

**Figure 4 biomedicines-08-00043-f004:**
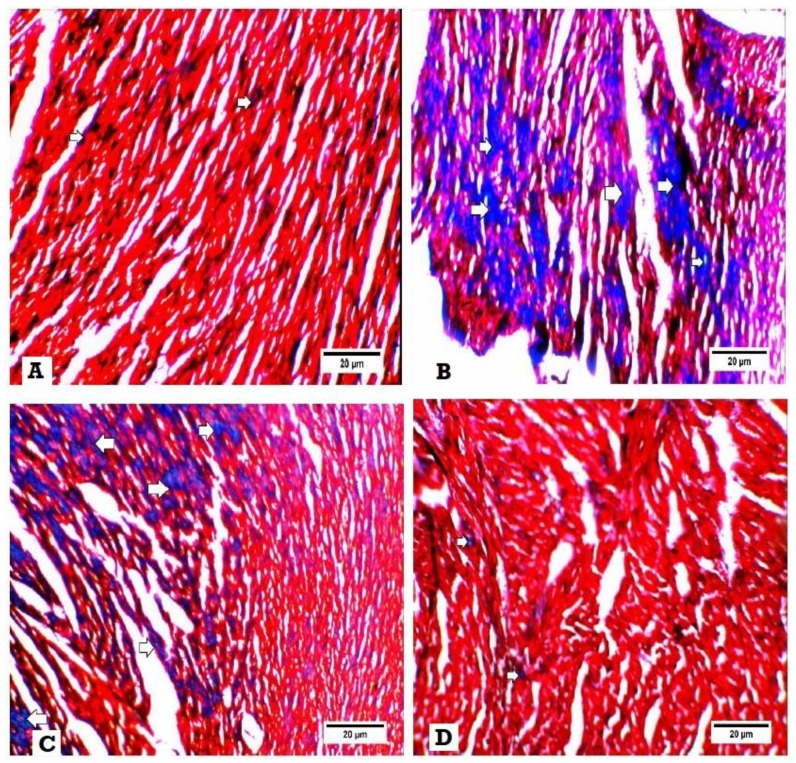
Histopathological examination for fibrosis by Masson trichrome. Interstitial fibrosis appears as blue colors. (**A**) Heart specimen from control group showing few interstitial fibrosis (400×), (**B**) heart specimen from the DM group showing marked interstitial fibrosis and collagen deposition (400×), (**C**) heart specimen from the DM + GLP1 group showing moderate interstitial fibrosis and collagen deposition (400x), and (**D**) heart specimen from the DM + SGLT2i group showing minimal interstitial fibrosis and collagen deposition (400×).

**Figure 5 biomedicines-08-00043-f005:**
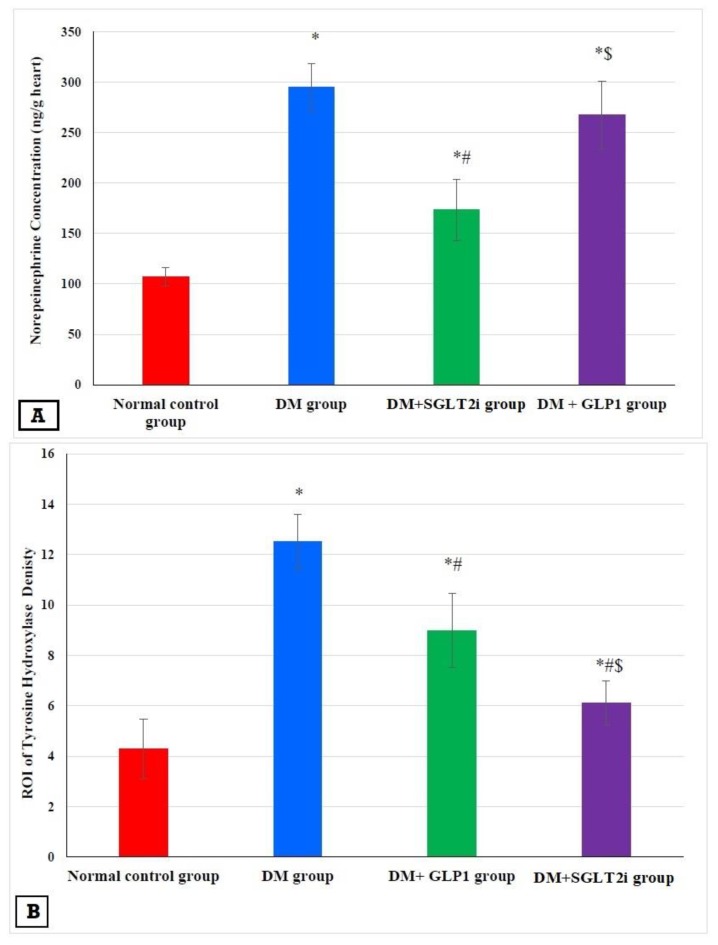
Effects of SGLTi and GLP1 on the myocardial norepinephrine content and density of sympathetic nerve fibers by tyrosine hydroxylase (TH). (**A**) The concentration of myocardial NE in different groups, (**B**) the density of sympathetic nerve fibers by TH immunostaining in different groups, (**C**) heart specimen from the control group showing evenly distributed TH-positive nerve fibers in the ventricular myocardium (arrows) (400×), (**D**) heart specimen from the DM group showing marked increase in immunopositivity for TH-positive fibers (arrows) (400×), (**E**) heart specimen from the DM +GLP1 group showing moderate increase in immunopositivity for TH-positive fibers (arrows) (400×), and (**F**) heart specimen from the DM +SGLT2i group showing minimal increase in TH-positive fibers (arrows) (400×). * Significant vs. control group, ^#^ significant vs. DM group, and ^$^ significant vs. DM + SGLT2i group.

**Figure 6 biomedicines-08-00043-f006:**
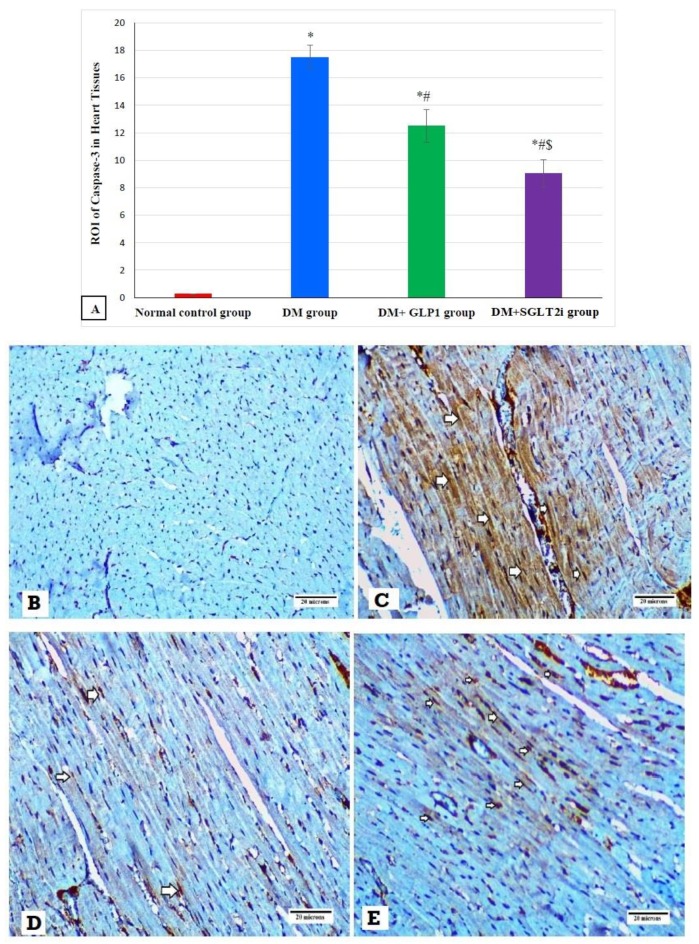
The expression of caspase-3 by immunostaining in different groups. (**A**) Graphs of ROI of the expression of caspase-3 in different groups. Heart specimens showing (**B**) negative cytoplasmic expression of caspase-3 in cardiomyocytes (control group), (**C**) marked cytoplasmic expression of caspase-3 in cardiomyocytes (arrows) (DM group), (**D**) moderate cytoplasmic expression of caspase-3 in cardiomyocytes (arrows) (DM + GLP1 group), and (**E**) minimal cytoplasmic expression of caspase-3 in cardiomyocytes (arrows) (DM + SGLT2i group). * Significant vs. control group, ^#^ significant vs. DM group, and ^$^ significant vs. DM + GLP1 group.

**Figure 7 biomedicines-08-00043-f007:**
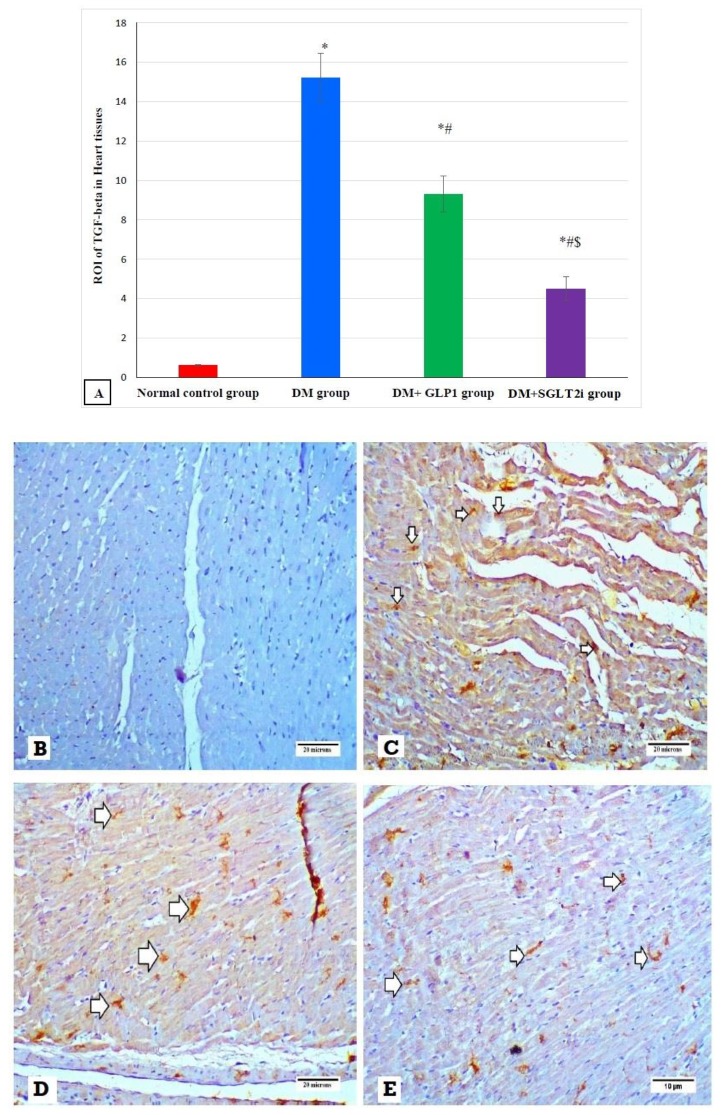
The expression of TGF-β by immunostaining in different groups. (**A**) Graphs of ROI of the expression of TGF-β in different groups. Heart specimens showing (**B**) negative TGF-β expression (control group), (**C**) marked membranous expression of TGF-β within the interstitial tissue between fibers (arrows) (DM group), (**D**) moderate membranous expression of TGF-β (arrows) (DM + GLP1 group), and (**E**) minimal membranous expression of TGF-β (arrows) (DM + SGLT2i group). * Significant vs. control group, ^#^ significant vs. DM group and ^$^ significant vs. GLP1 group.

**Table 1 biomedicines-08-00043-t001:** Effects of SGLT2i and GLP1 analogues on glucose homeostasis and cardiac enzymes in diabetic cardiomyopathy.

	Normal Control Group	Diabetes Mellitus (DM) Group	DM +SGLT2i Group	DM + GLP1 Group
Blood glucose (mg/dL)	91.50 ± 9.48	3698.83 ± 18.67 *	152.33 ± 9.627 *^#^	193.50 ± 9.39 *^#$^
Insulin (U/mL)	11.27 ± 0.29	6.78 ± 0.33 *	8.60 ± 0.70 *^#^	8.44 ± 0.68 *^#^
Homeostasis model assessment (HOMA) index	2.53 ± 0.16	6.14 ± 0.19 *	3.22 ± 0.31 *^#^	3.97 ± 0.28 *^#$^
LDH	250.33 ± 22.37	990.00 ± 56.21 *	292.67± 65.35 ^#^	296.33 ± 26.97 *^#^
CK-MB	20.67 ± 3.44	271.33 ± 16.73 *	80.17 ± 22.47 *^#^	161.00 ± 58.26 *^#^

All data are expressed as Mean ± SD. One-way ANOVA with Tukey Post hoc test. * Significant vs. normal control group, # significant vs. DM group, and $ significant vs. DM + SGLT2i group.

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
