# Peer review of "Comparative Study of the Effects of GLP1 Analog and SGLT2 Inhibitor against Diabetic Cardiomyopathy in Type 2 Diabetic Rats: Possible Underlying Mechanisms"

_biomedicines, 2020, doi:10.3390/biomedicines8030043_

Round 1
Reviewer 1 Report
Interesting paper focused on the effects of GLP1 analogue and SGLT2 inhibitor on DCM and diabetic CAN in T2DM and which clarify their underlying mechanisms.
Some comments may improve the manuscript:
re-organize the discussion and better explain the different interaction between (OS, inflammation, fibrosis etc.) in the incidence of DCM in T2DM population re-wrote the conclusion, English language might be improved include a paragraph showing if GLP1 and SGLT2 have some side effects I humans?
Author Response
Reviewer’s # 1
Comment
Interesting paper focused on the effects of GLP1 analogue and SGLT2 inhibitor on DCM and diabetic CAN in T2DM and which clarify their underlying mechanisms
Response
We would like to thank reviewer for this comment
Comment
Re-organize the discussion and better explain the different interaction between (OS, Inflammation, fibrosis etc.) in the incidence of DCM in T2DM population re-wrote the conclusion
Response
Done
Comment
English language editing might be improved include a paragraph showing if GLP1 and SGLT2 have some side effects on humans?
Response
English editing was done and corrections were written in red colour
Reviewer’s # 1
Comment
Interesting paper focused on the effects of GLP1 analogue and SGLT2 inhibitor on DCM and diabetic CAN in T2DM and which clarify their underlying mechanisms
Response
We would like to thank reviewer for this comment
Comment
Re-organize the discussion and better explain the different interaction between (OS, Inflammation, fibrosis etc.) in the incidence of DCM in T2DM population re-wrote the conclusion
Response
Done
Comment
English language editing might be improved include a paragraph showing if GLP1 and SGLT2 have some side effects on humans?
Response
English editing was done and corrections were written in red colour
Reviewer 2 Report
The manuscript entitled “Comparative Study of the Effects of GLP1 analog and SGLT2 inhibitor Against Diabetic Cardiomyopathy in Type 2 Diabetic Rats: Possible Underlying Mechanisms” by Hussein et al. demonstrates that an inhibitor of sodium-glucose cotransporter 2 (SGLT2) is more effective on alleviating damage of the heart than glucagon-like peptide 1 (GLP-1) in the streptozotocin-induced model of diabetes mellitus in rats. Their discussion is based on solid data and logic. The reviewer has several suggestions to improve the quality of this manuscript.
Most of readers may have some difficulty in understanding for what purpose the authors perform assay on tyrosine hydroxylase. It would be better to explain about it.
Page 4, line 12: “TH was quantified as …”
It seems that the description above corresponds to “ROI of Tyrosine Hydroxylase Density” appeared in Figure 5B. However, the relationship between the two is not clear.
Page 4, line 42: “DM+ SGLTi group showed significant reduction in Nrf2 expression …”
It seems that the authors did not measure the expression level of Nrf2 in this study.
Page 5, lime 18: “While hearts of DM group showed large number of TH positive small nerve twigs are scattered in a ‘snow-like’ pattern (Fig 7D).”
Readers may find difficulty in understanding this sentence.
Page 5, line 19: “snow-like pattern”
It is not obvious from Figure 5. Please explain.
Page 7, line 30: “he current showed significant increase in the density and number of sympathetic fibers in diabetic hearts with higher levels of NE in the myocardium suggesting sympathetic overactivity in heart tissues in early DM.”
Readers may find difficulty in understanding this sentence.
Page 7, line 36: “Moreover, we found in the current work significant reduction in TH density by either SGLTi or FLP1 with more powerful effect for SGLTi than GLP1 suggesting attenuation of the upregulated sympathetic nerve fibers is another possible mechanism for the cardioprotective effects of these new antidiabetic agents in DCM.”
Readers may find difficulty in understanding this sentence.
Author Response
Reviewer’s # 2
Comment
The manuscript entitled “Comparative Study of the Effects of GLP1 analog and SGLT2 inhibitor Against Diabetic Cardiomyopathy in Type 2 Diabetic Rats: Possible Underlying Mechanisms” by Hussein et al. demonstrates that an inhibitor of sodium-glucose cotransporter 2 (SGLT2) is more effective on alleviating damage of the heart than glucagon-like peptide 1 (GLP-1) in the streptozotocin-induced model of diabetes mellitus in rats. Their discussion is based on solid data and logic. The reviewer has several suggestions to improve the quality of this manuscript
Response
Thanks
Comment
Most of readers may have some difficulty in understanding for what purpose the authors perform assay on tyrosine hydroxylase. It would be better to explain about it
Response
Done in red colour in introduction and discussion
Comment
Page 4, line 12: “TH was quantified as….
It seems that the description above corresponds to “ROI of Tyrosine Hydroxylase Density” appeared in Figure 5B. However, the relationship between the two is not clear
Response
This point was clarified in methodology section (page 4, line 24 in red colour)
Comment
Page 4, line 42: “DM+ SGLTi group showed significant reduction in Nrf2 expression
It seems that the authors did not measure the expression level of Nrf2 in this study
Response
We would like to thank the reviewer for this comment. This was corrected to TNF-α expression as shown in revised manuscript
Comment
Page 5, lime 18: “While hearts of DM group showed large number of TH positive small nerve twigs are scattered in a ‘snow-like’ pattern (Fig 7D)
Readers may find difficulty in understanding this sentence
Response
This sentence was rewritten to be clear.
Comment
Page 5, line 19: “snow-like pattern
It is not obvious from Figure 5. Please explain
Response
This sentence was deleted in revised manuscript
Comment
Page 7, line 30: “the current showed significant increase in the density and number of sympathetic fibers in diabetic hearts with higher levels of NE in the myocardium suggesting sympathetic overactivity in heart tissues in early DM
Readers may find difficulty in understanding this sentence
Response
This sentence was clarified in revised manuscript
Comment
Page 7, line 36: “Moreover, we found in the current work significant reduction in TH density by either SGLTi or FLP1 with more powerful effect for SGLTi than GLP1 suggesting attenuation of the upregulated sympathetic nerve fibers is another possible mechanism for the cardioprotective effects of these new antidiabetic agents in DCM
Readers may find difficulty in understanding this sentence
Response
This sentence was clarified in revised manuscript